# Invasive *Wedelia trilobata* Performs Better Than Its Native Congener in Various Forms of Phosphorous in Different Growth Stages

**DOI:** 10.3390/plants12173051

**Published:** 2023-08-25

**Authors:** Die Hu, Irfan Ullah Khan, Jiahao Wang, Xinning Shi, Xinqi Jiang, Shanshan Qi, Zhicong Dai, Hanping Mao, Daolin Du

**Affiliations:** 1School of Agricultural Engineering, Jiangsu University, Zhenjiang 212013, China; hu_209die@163.com (D.H.); wjh980217@163.com (J.W.); j18280577017@163.com (X.J.); maohp@ujs.edu.cn (H.M.); 2Institute of Environment and Ecology, School of the Environment and Safety Engineering, Jiangsu University, Zhenjiang 212013, China; irfanullahkhan195@yahoo.com (I.U.K.); sxn13753938033@163.com (X.S.); daizhicong@163.com (Z.D.); ddl@ujs.edu.cn (D.D.); 3Jiangsu Collaborative Innovation Center of Technology and Material of Water Treatment, Suzhou University of Science and Technology, Suzhou 215009, China

**Keywords:** nutrient limited, resource fluctuations hypothesis, phosphorus absorption efficiency, invasion mechanism

## Abstract

At present, many hypotheses have been proposed to explain the mechanism of alien plants’ successful invasion; the resource fluctuations hypothesis indicates that nutrient availability is a main abiotic factor driving the invasion of alien plants. Higher phosphorus utilization and absorption efficiency might be one of the important reasons for alien plants successful invasion. *Wedelia trilobata*, one of the notorious invasive weeds in China, possesses a strong ability to continue their development under infertile habitats. In this study, firstly, *W. trilobata* and its native congener, *W. chinensis,* were grown in various phosphorus forms to test their absorption efficiency of phosphorus. Secondly, the different responses of *W. trilobata* and *W. chinensis* to the insoluble phosphorus in three growth stages (at 30, 60, and 150 days cultivation) were also tested. The results showed that the growth rate, root morphology, and phosphorus absorption efficiency of *W. trilobata* under various insoluble, organic, or low phosphorus conditions were significantly higher than that of *W. chinensis*. During the short-term cultivation period (30 d), the growth of *W. trilobata* under insoluble and low phosphorus treatments had no significant difference, and the growth of *W. trilobata* in insoluble phosphorus treatment also had no significant effect in long-term cultivation (60 and 150 d). However, the growth of *W. chinensis* in each period under the conditions of insoluble and low phosphorus was significantly inhibited throughout these three growth stages. Therefore, invasive *W. trilobata* had a higher phosphorus utilization efficiency than its native congener. This study could explain how invasive *W. trilobata* performs under nutrient-poor habitats, while also providing favorable evidence for the resource fluctuations hypothesis.

## 1. Introduction

Plant invasion can alter the structure and functions of ecosystems and cause both severe ecological problems and enormous economic losses [1]. Once exotic plants enter new habitats, they make various relationships with other plants and finally start competing with each other [2]. Many invasive plants have superior resource competitiveness traits, such as absorbing more ecological resources than native plants [3]. In previous studies, invasive and native plants were compared to explore their different responses to soil nutrients, especially with nitrogen and phosphorus [4,5]. The resource fluctuations hypothesis indicates that nutrient availability might be a major abiotic factor, which contributes to the successful invasion of alien plants [6]. It has been reported that invasive plants have a higher rate of resource utilization than native plants through their superior resource competitiveness [7]. Although exotic plants suffer from low-resource environments during their invasion, they can rapidly grow and expand even under nutrient-poor conditions, such as sand dunes, and arid and semi-arid grasslands [8]. Some invasive plants can make use of their own physiological and ecological advantages, are easy to quickly grow in the environment and have a strong ability to resist and remain tolerant to the adverse environment [9,10]. Therefore, studying how invasive plants respond to various nutrient forms could be important to understanding their intrinsic invasion mechanisms.

Essential nutrients are required for plant growth; in particular, nitrogen and phosphorus are very important constituents for plant growth and yielding [11]. Phosphorus (P) is one of the most important macro-nutrients which are required for plant growth and development. The primary source of P in terrestrial environments occurs from the weathering volcanic and atmospheric deposition [12]. There are two types of phosphorus present in the soil, such as organic P and inorganic P, and the most available form in the soil is organic P [13]. Although the total phosphorus content in the soil varies depending on the environment, the availability of phosphorus is extremely low in soils, and different soil types constitute different forms, such as in acidic soil, inorganic phosphorus mainly exists in the form of aluminum phosphate (Al-P) or iron phosphate (Fe-P), and in alkaline soil, inorganic phosphorus mainly exists in the form of calcium phosphate (Ca-P) [14], which usually have a low diffusion coefficient or high fixation rate in the soil [15]. It can be seen that the absorption and utilization of phosphorus by plants are very important for plants to survive and grow better in new environments. Although plant roots can absorb nutrients from the soil in different forms [16,17], phosphorus deficiency occurs regularly in the soil, and phosphorus is easily bound with Ca^2+^, Fe^3+^, Al^3+^, and other cations in soils to form insoluble phosphorus, which would have negative influences on photosynthesis, respiration, and biosynthesis for plants [18,19].

The absorption and utilization of phosphorus by plants can be determined using both the concentration gradient and the diffusivity of P in the soil near the roots. Efficient phosphorus utilization is the key to plant growth and development, especially in nutrient-limited habitats. Certainly, it is unlikely that a given adaptation can improve the access to multiple forms of insoluble P sources for a certain plant species [20]. A higher phosphorus absorption efficiency (PAE) is of great significance to enhance the plant adaptability in low-phosphorus habitats [21], and is also an important specialty for alien plants, which can invade nutrient-limited habitats successfully. Zhang et al. (2021) tested three different levels of phosphorus treatment; the invasive plants *Mikania micranatha* and *Chromolaena odorata* were found to have a higher phosphorus utilization efficiency under a low-phosphorus environment than their native plants [22].

*Wedelia trilobata* (Asteraceae), native to the tropics of South America, is one of the malignant invasive weeds in the world, widely distributed in southern China in the later parts of the 20th century [23]. Due to its extremely invasive nature, it grows very fast, has severely damaged the diversity and richness of local plant species, and caused huge economic losses [24]. In a previous field survey, we found that invasive *W. trilobata* was able to spread in an extremely nutrient-poor soil [24]. Based on this ecological phenomenon, we speculated that the invasive species *W. trilobata* has an efficient utilization of nutrient resources activity. Therefore, we hypothesized that invasive plants might utilize different forms of phosphorus and process a higher absorption efficiency of low phosphorus than its native species. This might be one of the crucial mechanisms that could explain its successful invasion in nutrient-limited habitats. The current study aimed to explore the differences in the growth of *W. trilobata* and its native congener *W. chinensis* under different phosphorus forms (including inorganic phosphorus, organic phosphorus, and low phosphorus) to elucidate the high-efficiency phosphorus utilization of the invasive plant. Additionally, we also explored the comparison between *W. trilobata* and native *W. chinensis* under the insoluble and low-soluble phosphorus under different cultured periods.

## 2. Results

### 2.1. Responses of Two Wedelia to Different Phosphorus Forms

Under different P forms, significantly different growth patterns of these two species were observed (Figure 1). The phenotypical result indicated that invasive *W. trilobata* grows better compared to *W. chinensis* under different P conditions. The leaf number and node number were not significantly different under the six phosphorus treatments for *W. trilobata* (Figure 1a,b). With the Ca-P and Al-P treatments, the shoot length and above-ground dry mass of *W. trilobata* were not significantly reduced (Figure 1c,d). On the other side, all the growth traits of native *W. chinensis*, including its leaf number, node number, shoot length, and above-ground day mass under insoluble phosphorus (Ca-P, Al-P, and Fe-P), organic phosphorus (O-P), and low-phosphorus (Low-P) treatments were significantly inhibited compared to the K-P treatments (Figure 1a–d).

Plant roots play important roles in the processing of plants absorbing and utilizing environmental resources. Root indicators, including the root length and root volume, could reflect the plant’s ability to acquire or absorb nutrients. For *W. trilobata*, their root length, root volume, and root dry mass were not significantly changed under the Ca-P, Al-P, and Low-P treatments compared with the K-P treatment, while they were significantly inhibited under the Fe-P and O-P conditions. For *W. chinensis*, their root length, root volume, and root dry mass were significantly reduced under different insoluble, organic, and low-phosphorus treatments (Figure 2a–d).

Compared with the K-P treatment, the nitrogen content of *W. trilobata* did not significantly differ among these phosphorus treatments, while the nitrogen content of *W. chinensis* was reduced in the Al-P and Fe-P treatments (Figure 3a). Compared with the K-P treatment, the phosphorus content of *W. trilobata* was significantly lower under the Al-P and Low-P treatments, while no significant differences were observed under other phosphorus treatments. The phosphorus content of *W. chinensis* under the Ca-P, Al-P, and Low-P treatments was significantly lower than under the K-P treatment (Figure 3b). The nitrogen–phosphorus ratio of *W. chinensis* was significantly higher than *W. trilobata* under Ca-P and Low-P conditions (Figure 3c). The PAE decrease ratio of *W. trilobata* was significantly lower compared to *W. chinensis* under the Ca-P and O-P treatments (Figure 3d).

### 2.2. Responses of Two Wedelia to Insoluble Phosphorus in Different Growth Stages

The responses of *W. trilobata* and *W. chinensis* were found to be significantly different in these three culture periods under phosphorus-limited conditions (Figure 4). There was no significant difference in the growth of *W. trilobata* under the Ca-P and Low-P treatments at 30 days of culture. At 60 days and 150 days of culture, the growth of *W. trilobata* under Ca-P was not significantly different from the control treatment K-P, while the leaf number, shoot length, and root number of *W. trilobata* under Low-P was inhibited. The root length was significantly increased at 60 days (Figure 4a–d). For native *W. chinensis*, their leaf number, shoot length, root number, and root length under the Ca-P and Low-P treatments were significantly inhibited at 30, 60, and 150 days of culture. However, at 60 days of culture, the root number of *W. chinensis* under the Ca-P culture significantly increased compared with K-P, and there was no significant change in the root length among treatments. At 150 days of incubation, the leaf number and shoot length of *W. chinensis* under the three phosphorus treatments were more consistent with those at 30 days; the growth was significantly inhibited under Ca-P and Low-P nutrition, but there were no significant differences in the root length and root number (Figure 4e–h).

At 30 days of incubation, there was no significant difference on the above- and below-ground dry mass of *W. trilobata* under the Ca-P and Low-P treatments. The above- and below-ground growth of *W. trilobata* under Ca-P had no significant difference at 60 and 150 days of culture. Under Low-P treatment, the above-ground dry mass of *W. trilobata* at 60 days and 150 days, and also below-ground dry mass at 150 days, were significantly lower compared to the K-P treatment (Figure 5a,b). However, under these three growth stages, the Ca-P and Low-P treatments significantly inhibited the above- and below-ground growth of *W. chinensis*, except for the below-ground dry mass at 60 days (Figure 5c,d). In the early growth stage (30 days), the phosphorus content of *W. trilobata* was not significantly correlated with its biomass, while it was significantly correlated with the total biomass of *W. chinensis* (Figure 6a). The total biomass of both *Wedelia* species showed a significant positive correlation with P content at 60 days (Figure 6b) and 150 days (Figure 6c). Compared to the K-P treatment, *W. trilobata* had a lower rate of PAE reduction than *W. chinensis* under the Ca-P and Low-P treatments at 30 days and 60 days, respectively (Figure 6d).

## 3. Discussion

The current study sought out to achieve its aim by conducting two experiments under different phosphorus conditions and under three growth stages. The first experiment using sand culture and hydroponics culture in the second experiment are quite different from the natural soil condition. The sand culture is closer to the natural soil condition as it provides ventilation and a solid rooting medium that is easier for plants to grow [25]. However, the use of a hydroponics culture can reduce the extent of mechanical damage to the root tissue when separating it and can more precisely control the availability of nutrients [26]. Therefore, a sand culture was adopted in the first experiment to facilitate the rooting of plants and be closer to the natural growth state of plants, while a hydroponics culture was adopted in the second experiment to facilitate the absorption of nutrients by plants and more accurately control the availability of plant nutrients. The results showed that invasive *W. trilobata* grows better under the different forms of phosphorus than its native congener *W. chinensis*. Also, *W. trilobata* has a higher calcium phosphorus utilization efficiency. Low phosphorus did not significantly inhibit the growth of invasive *W. trilobata* during early-stage growth. However, low phosphorus treatment significantly inhibited all three growth stages of native *W. chinensis*.

The first experiment observed that the growth of invasive *W. trilobata* was better compared to native *W. chinensis* under the different forms of phosphorus. This result indicated that *W. trilobata* could absorb or utilize different forms of insoluble phosphorus and low-phosphorus resources better than native *W. chinensis*. Some studies have shown that invasive plants may have better innate advantages than native plants, which are derived from their physiological and ecological advantages, including having better resource acquisition strategies, and the ability to obtain higher resource acquisition rates [27,28]. Even under low-nutrient environments, invasive species appear to be more efficient in the uptake and utilization of nutrients than native species, which is reliable with our current results [3]. It has been reported that some invasive plants can be disturbed under low-nutrient environments [29]. A higher and efficient utilization of nutrients may be the essential mechanism to explain why invasive plants have a strong capability to adjust to poor-nutrient habitats.

The plant primary response to the soil nutrient status is the root system [29], and plants can respond to insoluble phosphorus and low-phosphorus environments by changing their root morphology, topology, or distribution patterns [30]. The current results indicated that the root morphology of invasive *Wedelia* were not significantly decreased under Ca-P, Al-P, and Low-P conditions (Figure 2), which may help invasive plants to improve their ability to absorb and utilize insoluble phosphorus and low-phosphorus forms. Furthermore, some studies have reported that the root system can also increase the availability of phosphorus in soil solutions by secreting H^+^/OH^−^, organic acids, or phosphatases [31,32], and also reported that root exudates could alter the rhizosphere environment and biological properties to improve the availability of nutrients and promote plant growth. Funk et al. (2013) also experimentally proved that the growth of invasive *Solidago canadensis* was better under insoluble or low P conditions compared to native species; similarly, the number of roots was significantly increased [3]. Therefore, it was highly possible for invasive species to expand the foraging range of P nutrition by increasing the number of roots, so as to improve the ability of invasive species to adjust their resource allocation under environmental stress. Therefore, the high phosphorus utilization and uptake capacity of invasive weeds may be explained more clearly from the aspect of the root system.

The N:P ratio could reflect the responses of plants to nutrient-limiting factors [30]. Our results showed that the N:P ratio of invasive *Wedelia* under the Ca-P and Low-P treatments were significantly higher than that of native *Wedelia*, indicating that the growth of invasive *Wedelia* was not restricted by Ca-P and Low-P (Figure 3c). This might be due to the phosphorus adaptation strategy of invasive *Wedelia*, which changed the bioavailability of Ca-P and possessed a high P absorption capacity. It has been reported that *Arabidopsis* has evolved systematic mechanisms to respond extremely well to low-phosphorus stress [33]. More molecular studies on invasive weeds response to low P are needed.

In the second experiment, the growth of invasive *Wedelia* were not significantly changed under insoluble or low P forms during their early growth stage, while the growth of native *Wedelia* was significantly inhibited (Figure 4 and Figure 5). Invasive *Wedelia* has a higher low-phosphorus and insoluble phosphorus utilization capacity than its native congener during its early stage of growth. It has been reported that different species may vary in their nutrient requirements [34]. In the early growth stage, invasive *Wedelia* may have less demands for phosphorus than native *Wedelia*, and these innate advantages may have great significance for invaders at the beginning of their invasion, especially for the invasion of nutrient-limited habitats. After 60 days of culture, there was no significant difference in the growth of invasive *Wedelia* under insoluble P. That is, invasive *Wedelia* was also able to utilize the insoluble phosphorus to meet their demand for phosphorus. However, for native *Wedelia*, their growth was significantly inhibited under both insoluble and low P conditions, especially under the low-P condition; this may be due to the limitation of the total amount of phosphorus and their insufficient ability to utilize the insoluble phosphorus, which inhibits growth. The phosphorus absorption efficiency (PAE) is very useful for providing plants with a high phosphorus efficiency [30]. It has been reported that the correlation analysis between plant P content and plant biomass showed that the phosphorus absorption content was positively correlated with the plant’s growth [35,36]. Our results showed that invasive *Wedelia* had a lower PAE decrease ratio than native *Wedelia* under various P forms and growth stages. This means that the PAE of invasive *Wedelia* were not greatly decreased under different insoluble or low-P forms. Invasive *Wedelia* had a higher PAE than native *Wedelia*. It has been reported that due to having a higher PAE, invasive species possess a higher development rate under low-nutrient environments [37]. Hu et al. (2018) confirmed that compared with natives, the phosphorus utilization capacity of invasive plants was higher and elevated with the increase in the invasion degree under the same soil phosphorus conditions [38]. Thorpe et al. (2006) used a combination of field and greenhouse experiments to suggest that *Centaurea maculosa* possessed a higher phosphorus absorption efficiency than the native species [39], which is consistent with our results. Our results were correlated with several studies, in which the invasive species growth rate was higher under very low-nutrient environments [40,41].

## 4. Materials and Methods

### 4.1. Plant Species

We assessed the responses to different phosphorus forms on a pair of invasive and *native Wedelia*. Invasive *W. trilobata* was collected from Xiamen City, Fujian Province, China. Native *W. chinensis* was collected from Guangzhou City, Guangdong Province, China [10,42]. Both plants were propagated in a greenhouse at Jiangsu University (119°31.76′ E, 32°12.02′ N). In the current study, healthy plant stem segments with two nodes of *W. trilobata* and *W. chinensis* were chosen for further experiments.

### 4.2. Experimental Setup for Growth in Different Phosphorus Forms

In this experiment, six different phosphorus forms were used, including three insoluble inorganic phosphorus: (1) hydroxyapatite (Ca_5_(PO_4_)_3_(OH), denoted Ca-P); (2) aluminum phosphate (AlPO_4_, denoted Al-P); (3) iron phosphate (FePO_4_, denoted Fe-P); one insoluble organic phosphorus: (4) phytic acid (C_6_H_18_O_24_P_6_, denoted O-P) and two soluble phosphorus: (5) potassium dihydrogen phosphate with a normal nutrient concentration (KH_2_PO_4_, K-P, as the control treatment); and a (6) low concentration of potassium dihydrogen phosphate (KH_2_PO_4_, Low-P). We used a phosphorus content of 30 mg/Kg for the different forms of phosphorus treatment except for the Low-P treatment; the phosphorus content for the Low-P treatment was 0.3 mg/Kg. The soluble phosphorus source was added in the form of a solution, and the insoluble phosphorus source was added in the form of powder and thoroughly mixed with sand. Next, 100 g of autoclaved sand, which was washed thoroughly with deionized water to remove nutrients, with different forms of P was put into flower pots (6.4 × 6.4 × 5.2 cm). Except for the K-P treatment, all other phosphorus treatments were supplemented with potassium chloride to ensure that all plants received the same content of potassium. Stems of *W. trilobata* or *W. chinensis* were inserted into the center of the flower pots. Five replicates were set up for each treatment with a total of 60 pots (6 treatments × 2 species × 5 replicates). For each pot, 25 mL of modified 0.5 × Hoagland nutrient solution without phosphorus was added to the sand every week to meet the plant’s nutritional needs. Purified water was added to all the plants when needed. All the treatments were cultivated in the greenhouse at Jiangsu University. The plants were harvested after 2 months and their growth characteristics, including their leaf number, node number, shoot length, and above-ground and root biomass, root morphological indicators, and physiological index, including their N and P contents, were all measured. The N:P ratio (the ratio of nitrogen content and phosphorus content) was also calculated in this study. The phosphorus absorption efficiency (PAE) was defined as total phosphorus content in the plant (mg plant^−1^) [43]. In this study, the PAE of the root (the ratio of total P content and root dry biomass) was calculated to assess the absorbed phosphorus ability of the plant roots. To better compare the differences in the PAE between these two species, the PAE decrease ratio (%) for each treatment was also calculated based on their PAE values using the following equation:PAE decrease ratio (%) = (PAE _treatment_ − PAE _control_)/PAE _control_
where PAE _treatment_ is the PAE value of the different phosphorus treatments of Ca-P, Al-P, Fe-P, O-P, or Low-P, and PAE _control_ is the PAE value of the K-P treatment.

### 4.3. Experimental Setup for Growth in Insoluble Phosphorus under Different Growth Stages

Based on the first experiment results, we addressed a question: Does the invasive *Wedelia* always perform well in phosphorus-limited media under all growth stages? According to the first experiment results, *W. trilobata* had a higher calcium phosphorus utilization efficiency than the native plants. We chose calcium phosphorus to test the different responses of these two species under three growth stages using a hydroponic experiment. Three kinds of phosphorus nutrition conditions were added into a 0.5 × phosphorus-free Hoagland nutrient solution in this experiment: (1) normal phosphorus (KH_2_PO_4_; K-P), (2) insoluble phosphorus (Ca_5_(PO_4_)_3_(OH); Ca-P), and (3) low phosphorus (KH_2_PO_4_; Low-P). The phosphorus content of K-P and Ca-P was 30 mg/L, while the phosphorus content of Low-P was 0.3 mg/L. Except for the K-P treatment, potassium chloride was added into the other two treatments to ensure that all plants received the same content of potassium. For the growth stages, we harvested plants under three growth stages, including the short-term period (30 d), mid-term period (60 d), and long-term period (150 d). There were a total of 126 flasks (3 phosphorus treatments × 3 growth stages × 2 species × 7 replicates) with each plant per flask (2.4 × 6.5 × 10 cm). All plants were cultured in an incubator with an appropriate environment (temperature: 26 ± 2 °C; photoperiod: 16h/8h (day/night); and light intensity: 450 μmol m^−2^ s^−1^). The placement of each flask was randomly adjusted weekly to avoid possible environmental differences that could affect the experimental results. At different harvesting times, we measured the leaf number, shoot length, root number, root length, and the above- and below-ground dry mass of the plant. The WinRHIZO root analysis system was used to analyze the effects of different forms of phosphorus sources on the root structure. The correlation between phosphorus uptake and total biomass was then calculated.

### 4.4. Determination of Nitrogen and Phosphorus Content in Plants

All the plants were ground and analyzed for total N content [44] and P content [45]. The nitrogen content was measured using the Kjeldahl digestion method (digestion for 1h at 200 °C and 2 h at 340 °C in a mixture of concentrated sulfuric acid and 30% hydrogen peroxide). After digestion, the plants were cooled and diluted with deionized water. The P content was determined calorimetrically using a UV-1200 spectrophotometer (MAPADA, Shanghai, China) [46].

### 4.5. Data Processing and Statistical Analysis

SPSS 25.0 software (IBM Corp., Armon, NY, USA) was used to perform a one-way ANOVA and multiple comparisons test (Duncan’s test, *p* < 0.05), which analyzed the effects of various phosphorus treatments on the growth of the two species and also under the different culture periods. Results of the statistical analysis of the data were plotted using Origin 2018 software (Origin Lab Co., Northampton, MA, USA).

## 5. Conclusions

In conclusion, all the results of this study showed that under the condition of phosphorus nutrient limitation, invasive *Wedelia* was more tolerant against the low-nutrient condition and has a higher PAE compared to the native species. The higher PAE may contribute to the growth advantage of invasive *Wedelia*. It can make better use of various phosphorus resources to promote its growth and rapid expansion under new habitats. Invasive species may succeed under low-resource environments by employing their resource conservation traits, such as their high resource-use efficiency [47]. It has also been reported that due to the low soil P content, the characteristics of invasive *Wedelia*, under the terms of phenotypical, physiological, and molecular, was better compared to other species [29]. Our results also provide favorable evidence for the resource fluctuations hypothesis. This may contribute to explain its successful invasion, particularly under nutrient-limited habitats. Furthermore, how invasive *Wedelia* utilize phosphorus resources efficiently is still unclear, and these findings will be required for further research.

## Figures and Tables

**Figure 1 plants-12-03051-f001:**
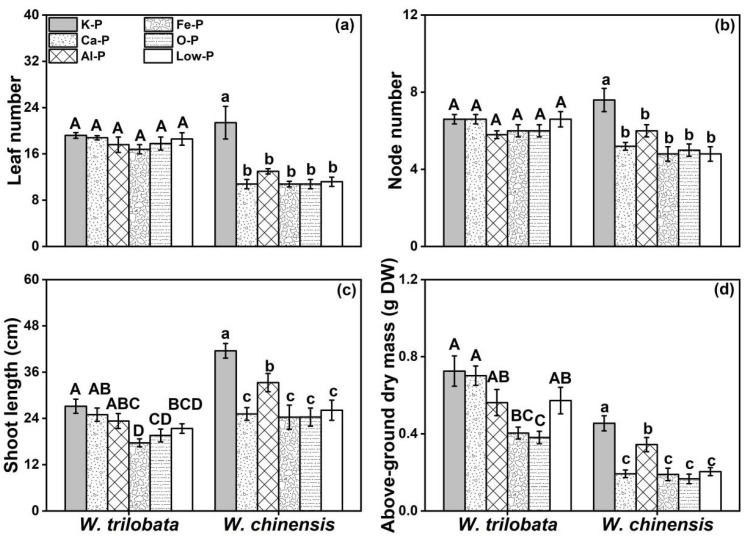
Effects of different phosphorus treatments on the (**a**) leaf number, (**b**) node number, (**c**) shoot length, and (**d**) above-ground dry mass of *W. trilobata* and *W. chinensis*. K-P: P was added as KH_2_PO_4_; Ca-P: P was added as Ca_5_ (PO_4_)_3_ (OH); Al-P: P was added as AlPO_4_; Fe-P: P was added as FePO_4_; O-P: P was added as C_6_H_18_O_24_P_6_; and Low-P: P was added as a low concentration of KH_2_PO_4_. Different letters indicate significant differences between treatments using Duncan’s test (*p* < 0.05), with uppercase letters for *W. trilobata* and lowercase letters for *W. chinensis*. Mean values and standard errors are shown (*n* = 5).

**Figure 2 plants-12-03051-f002:**
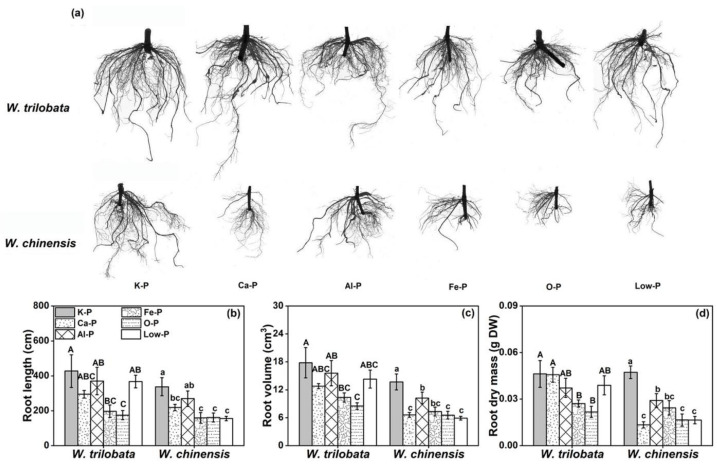
Effects of different phosphorus treatments on the (**a**) root morphology, (**b**) root length, (**c**) root volume, and (**d**) root dry mass of *W. trilobata* and *W. chinensis*. K-P: P was added as KH_2_PO_4_; Ca-P: P was added as Ca_5_ (PO_4_)_3_ (OH); Al-P: P was added as AlPO_4_; Fe-P: P was added as FePO_4_; O-P: P was added as C_6_H_18_O_24_P_6_; and Low-P: P was added as a low concentration of KH_2_PO_4_. Different letters indicate significant differences between treatments using Duncan’s test (*p* < 0.05), with uppercase letters for *W. trilobata* and lowercase letters for *W. chinensis*. Mean values and standard errors are shown (*n* = 5).

**Figure 3 plants-12-03051-f003:**
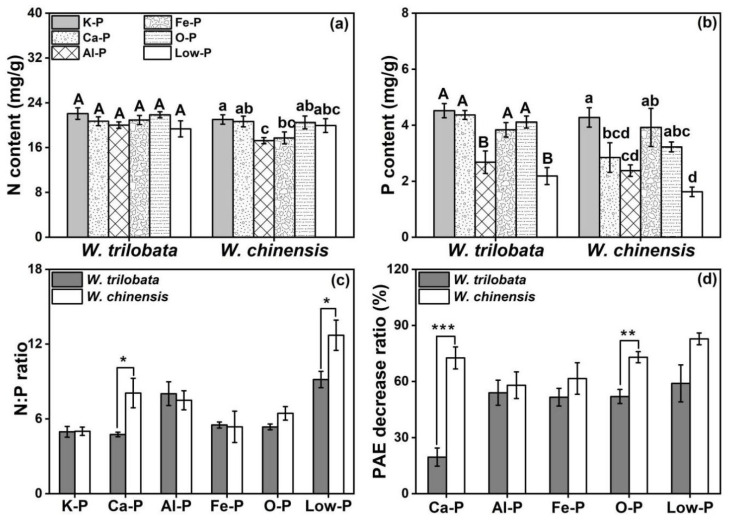
Effects of different phosphorus treatments on (**a**) nitrogen content, (**b**) phosphorus content, (**c**) nitrogen–phosphorus ratio, and (**d**) PAE decrease ratio of *W. trilobata* and *W. chinensis*. K-P: P was added as KH_2_PO_4_; Ca-P: P was added as Ca_5_ (PO_4_)_3_ (OH); Al-P: P was added as AlPO_4_; Fe-P: P was added as FePO_4_; O-P: P was added as C_6_H_18_O_24_P_6_; and Low-P: P was added as a low concentration of KH_2_PO_4_. Different letters indicate significant differences between treatments using Duncan’s test (*p* < 0.05), with uppercase letters for *W. trilobata* and lowercase letters for *W. chinensis.* Nitrogen–phosphorus ratio and PAE decrease ratio using the *t*-test; * represents *p* < 0.05, ** represents *p* < 0.01, and *** represents *p* < 0.001. Mean values and standard errors are shown (*n* = 5).

**Figure 4 plants-12-03051-f004:**
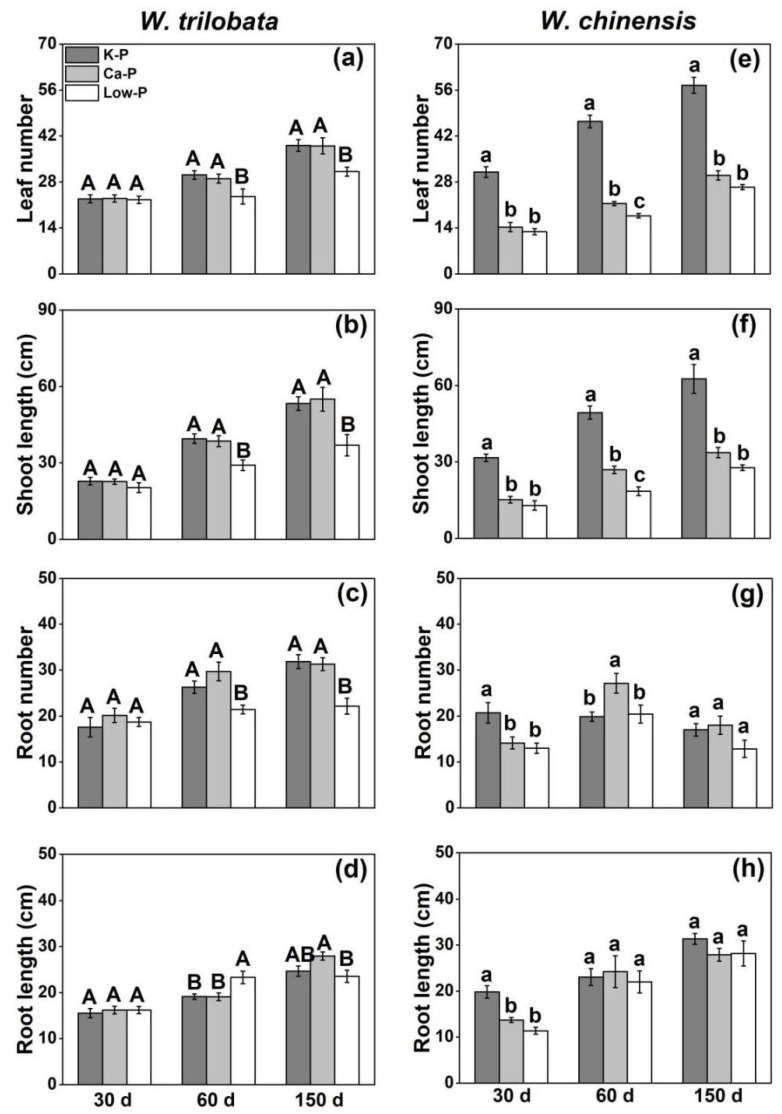
Effects of different phosphorus treatments on the (**a**,**e**) leaf number, (**b**,**f**) shoot length, (**c**,**g**) root number, and (**d**,**h**) root length of *W. trilobata* and *W. chinensis* at different periods. K-P: P was added as KH_2_PO_4_; Ca-P: P was added as Ca_5_(PO_4_)_3_(OH), and Low-P: P was added as a low concentration of KH_2_PO_4_. Different letters indicate significant differences between treatments using Duncan’s test (*p* < 0.05), with uppercase letters for *W. trilobata* and lowercase letters for *W. chinensis*. Mean values and standard errors are shown (*n* = 7).

**Figure 5 plants-12-03051-f005:**
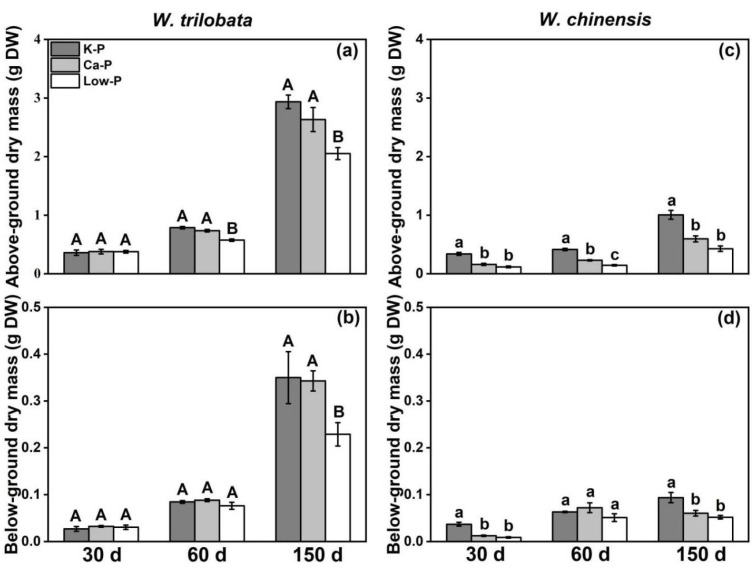
Effects of different phosphorus treatments under different periods on the (**a**,**c**) above-ground and (**b**,**d**) below-ground dry mass of *W. trilobata* and *W. chinensis*. K-P: P was added as KH_2_PO_4_; Ca-P: P was added as Ca_5_(PO_4_)_3_(OH), and Low-P: P was added as a low concentration of KH_2_PO_4_. Different letters indicate significant differences between treatments using Duncan’s test (*p* < 0.05), with uppercase letters for *W. trilobata* and lowercase letters for *W. chinensis*. Mean values and standard errors are shown (*n* = 7).

**Figure 6 plants-12-03051-f006:**
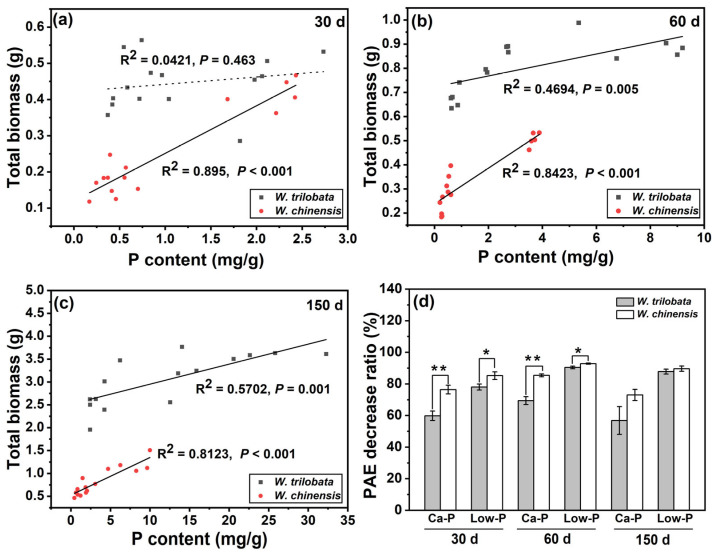
The correlation between P content and plant total biomass under different phosphorus treatments at (**a**) 30 days of growth, (**b**) 60 days of growth, and (**c**) 150 days of growth, and (**d**) the PAE decrease ratio of *W. trilobata* and *W. chinensis*. * represents *p* < 0.05; ** represents *p* < 0.01. Mean values and standard errors are shown (*n* = 7).

## Data Availability

The data presented in this study are available on request from the corresponding author (e-mail: qishanshan1986120@163.com).

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
