# Peer review of "Invasive Wedelia trilobata Performs Better Than Its Native Congener in Various Forms of Phosphorous in Different Growth Stages"

_plants, 2023, doi:10.3390/plants12173051_

Round 1

Reviewer 1 Report

The introduction lacks the research progress of phosphorus absorption efficiency (PAE) in different forms by plants (including invasive plants and native plants), as well as the previous studies of phosphorus absorption efficiency (PAE) of invasive plants. This makes the hypothesis in this study (Line 81-82) is somewhat abrupt.

Line 265: “Guangzhou Province” is wrong. I guess it should be “Guangdong Province”.

Line 270: Why did you set the six P treatments?

Line 293-298: I cannot understand PAE and PAE decrease ratio. Furthermore, the cited reference 34 is incorrect, where there was no PAE definition and calculation.

There was no plant biomass of the other 3 P treatments in figure 4 & 5 as the figures 1-3. Why?

It is unreasonable to correlate P uptake with biomass because P uptake is calculated as P concentration multiply by biomass. P uptake has included biomass. Why did you do this correlation analysis in figure 6? In addition, what is the basis for calculating PAE? Thus, the corresponding results, discussion and conclusion should be rewritten, including the conclusion in the abstract.

Author Response

We really appreciated the reviewers valuable comments to improve our manuscript. We have revised the whole text according to the reviewers comments as follows:

Point 1: The introduction lacks the research progress of phosphorus absorption efficiency (PAE) in different forms by plants (including invasive plants and native plants), as well as the previous studies of phosphorus absorption efficiency (PAE) of invasive plants. This makes the hypothesis in this study (Line 81-82) is somewhat abrupt.

Response 1: The research progress of phosphorus absorption efficiency (PAE) of different forms by plants (including invasive plants and native plants), as well as previous studies of phosphorus absorption efficiency (PAE) by invasive plants, have been included in the introduction (Line 76-82).

Point 2: Line 265: “Guangzhou Province” is wrong. I guess it should be “Guangdong Province”.

Response 2: It has been modified to "Guangdong Province" (Line 279).

Point 3: Line 270: Why did you set the six P treatments?

Response 3: There are organic and inorganic phosphorus in natural soil. We set one form of organic phosphorus (O-P). Inorganic phosphorus mainly exists in these three forms: aluminum phosphate (Al-P), iron phosphate (Fe-P) and calcium phosphate (Ca-P). Since these three kinds of phosphorus are insoluble and difficult to be absorbed by plants, we set up soluble phosphorus treatment as control treatment which is easily absorbed by plants (K-P). In other hand, we also assumed that if invasive plant could use phosphorus more efficiently in low content phosphorus. Therefore, we also set up low phosphorus concentration treatment (Low-P).

Point 4: Line 293-298: I cannot understand PAE and PAE decrease ratio. Furthermore, the cited reference 34 is incorrect, where there was no PAE definition and calculation.

Response 4: PAE was defined as total phosphorus content in the plant (mg plant-1), and PAE decrease ratio referred to the proportion of PAE values of different treatment and the control treatment (K-P). The value of PAE decrease ratio is better to compare the significant differences between the invasive and native Wedelia. We have made this clear in the text (Line 308-315). In addition, the incorrect citation 34 have been replaced to citation 39.

Point 5: There was no plant biomass of the other 3 P treatments in figure 4 & 5 as the figures 1-3. Why?

Response 5: In this study, we conducted two experiments. The first experiment is to test the different response of two species in the six different phosphorus forms (the results are presented in figures 1-3). Then, in the second experiment, we chose only one insoluble treatment (Ca-P) among those six treatments to test the different responses to insoluble phosphorus of these two species in different growth stages (the results are presented in figures 4-6). Therefore, in Figure 4-6, we focus on the different responses of these two species in three growth stages (30d, 60d, and 150d) instead of different phosphorus forms. We have made this clear in the Materials and Methods part (Line 317-321)

Point 6: It is unreasonable to correlate P uptake with biomass because P uptake is calculated as P concentration multiply by biomass. P uptake has included biomass. Why did you do this correlation analysis in figure 6? In addition, what is the basis for calculating PAE? Thus, the corresponding results, discussion and conclusion should be rewritten, including the conclusion in the abstract.

Response 6: It is reported that the absorption of nutrient elements by plants could affect plant biomass (Wang et al. 2017). Therefore, we tried to assess the correlation of P uptake and the plant biomass. The results showed that there was no significant correlation between the P uptake and the biomass of invasive Wedelia, while there was a significant correlation between native Wedelia, indicating that the invasive plant was more tolerant to low nutrient conditions. As for the calculation of PAE, we referred to Douglas et al. (2022). We also state about the PAE in the text (Line 16-17 in abstract, Line 76-82 in introduction, Line 262-268 in discussion, and Line 356-357 in conclusion).

Wang, J.; Liu, P.; Liu Z.Y.; Wu Z.S; Li, Y.S.; Guan, X.Y. Dry matter accumulation and phosphorus efficiency response of cotton cultivars to phosphorus and drought. Journal of Plant Nutrition 2017, 40, 2349-57.

Douglas José Marques, Ernani Clarete da Silva, José Andrés Carreño Siqueira, Elham Abedi, Fernanda Rosa Veloso, Gabriel Mascarenhas Maciel, Wilson Roberto Maluf. Variation in the dynamic of absorption and efficiency of phosphorus use in tomato. Sci Rep. 2022. 14: 4379.

Reviewer 2 Report

The article deals with an interesting issue of the competitive advantage of invasive species, resulting from their broader ecological scale and the ability to use environmental resources more effectively. The text is well-written, easy to read. I found no significant bugs.

Author Response

Response: Thank you very much for your recognition on this study. We have edited the whole text according to the other two reviewers to improve this manuscript.

Reviewer 3 Report

The manuscript entitled "Invasive Wedelia trilobata performs better than native congener in various forms of phosphorous in different growth stages" elucidating how the invasive W. trilobata performances in nutrient-poor habitats using greenhouse and hydroponic experiments. The article is interesting and contains valuable results. The followings are my specific comments:

1. Actually, previous studies mainly focused on the responses of invasive and native species to nitrogen addition/nitrogen deposition (e.g. Guo et al 2023, Ecotoxicology and Environmental Safety; Wang et al, 2022, STOTEN). The current study, focusing on the invasive Wedelia trilobata and its native congener responding to various forms of phosphorous, adds some new insights into this research area. I would suggest the authors demonstrate this in the introduction.

2. It should be also explained that how many stem segments were used in the second experiment, how many stem segments were planted per pot, and the size of the pots.

3. Please state the reason why you choose calcium phosphorus in the second experiment since you used many forms of phosphorus in the first experiment.

4. Please give more explanations in discussion part that why the growth of W. trilobata under insoluble and low phosphorus treatments had no significant difference during the short-term cultivation (30 d)? And what is the ecological significant for the invader W. trilobata ?

Author Response

We really appreciated the reviewers valuable comments to improve our manuscript. We have revised the whole text according to the reviewers comments as follows:

Point 1: Actually, previous studies mainly focused on the responses of invasive and native species to nitrogen addition/nitrogen deposition (e.g. Guo et al 2023, Ecotoxicology and Environmental Safety; Wang et al, 2022, STOTEN). The current study, focusing on the invasive Wedelia trilobata and its native congener responding to various forms of phosphorous, adds some new insights into this research area. I would suggest the authors demonstrate this in the introduction.

Response 1: Thank you for the suggestion. We added these studies in the introduction part.

Point 2: It should be also explained that how many stem segments were used in the second experiment, how many stem segments were planted per pot, and the size of the pots.

Response 2: In the second experiment, there were three different phosphorus treatments and three growth stages for the two plant species and 7 replications for each treatment, therefore, there are in total 126 plant stem segments were used. We also gave more details for the planting in Materials and Methods part (Line 329-331).

Point 3: Please state the reason why you choose calcium phosphorus in the second experiment since you used many forms of phosphorus in the first experiment.

Response 3: In the first experiment, we found that the PAE decrease ratio of W. trilobata and W. chinensis under Ca-P treatment were significantly different, which was not seen under other phosphorus treatments. That was to say, W. trilobata had higher calcium phosphorus utilization efficiency. Therefore, we chose calcium phosphorus in the second experiment. We have stated this in Materials and Methods part (Line 317-321).

Point 4: Please give more explanations in discussion part that why the growth of W. trilobata under insoluble and low phosphorus treatments had no significant difference during the short-term cultivation (30 d)? And what is the ecological significant for the invader W. trilobata ?

Response 4: More explanations have been given in the discussion part (Line 252-256).

Round 2

Reviewer 1 Report

Basically, I’m satisfied with the revision. The present study using sand culture (the first experiment) and hydraulic culture (the second experiment) are quite different from the natural soil condition. The authors should discuss this difference and such limitation of the present study. In addition, I still think the authors should point out the bias to correlate P uptake with biomass (regarding to Fig. 6) because P uptake is calculated as P concentration multiply by biomass. The authors should provide the necessity and reason to do this correlation analysis in figure 6.

Author Response

We really appreciated the reviewers valuable comments to improve our manuscript. We have revised the whole text according to the reviewers comments as follows:

 Point 1: Basically, I’m satisfied with the revision. The present study using sand culture (the first experiment) and hydraulic culture (the second experiment) are quite different from the natural soil condition. The authors should discuss this difference and such limitation of the present study.

 Response 1: Thank you for the suggestion. It is right that soil condition could be more consistent to natural condition. However, the nutrient in natural soil is not easy to be controlled compared to sand and hydroponics culture. In order to test the response of native and invasive species to different P forms, we need to precisely control the P content in culture medium. Therefore, sand without nutrient and hydraulic culture would be better choice.

We got six different P forms in the first experiment including soluble, insoluble and organic phosphorus. Sand medium is more approximate to soil medium, all the different P forms could better exist in sand medium. In the second, we need to assess more accurate root growth in the insufficient P condition, therefore, we use hydroponics culture to reduce mechanical damage to root tissue (Yugandhar et al., 2017). It is easier to observe root growth and more precise nutrient control in hydroponics culture (Nguyen et al., 2016).

Nevertheless, we have stated the differences and the limitation of these two culture methods in the discussion part (Line 205-214).       

Yugandhar, P.; Veronica, N.; Panigrahy, M.; Nageswara Rao, D.; Subrahmanyam, D.; Voleti, S.R.; Mangrauthia, S.K.; Sharma, R.P.; Sarla, N. Comparing hydroponics, sand, and soil medium to evaluate contrasting rice nagina 22 mutants for tolerance to phosphorus deficiency. Crop Sci 2017, 57, 2089-97. https://doi.org/10.2135/cropsci2016.07.0594.

Nguyen, N.T.; McInturf, S.A.; Mendoza-Cózatl, D.G. Hydroponics: a versatile system to study nutrient allocation and plant responses to nutrient availability and exposure to toxic elements. J Vis Exp 2016, 113, e54317. https://doi.org/10.3791/54317.

Point 2:  In addition, I still think the authors should point out the bias to correlate P uptake with biomass (regarding to Fig. 6) because P uptake is calculated as P concentration multiply by biomass. The authors should provide the necessity and reason to do this correlation analysis in figure 6.

Response 2: Thank you for pointing out this problem. Here is our mistake to make you confused on this figure. We would like to assess the relationship of P content and biomass. The unit of P content is mg/g, which is the P content of per gram of plants. We have edited this figure. It is reported that the correlation analysis between plant P content and plant biomass showed that the phosphorus absorption content was positively correlated with plant growth (Zhao et al. 2021 ;Yang et al. 2021 ). Therefore, we attempted to assess the correlation between P content and the plant biomass. The results showed that the correlation and slope between the P content and the biomass of invasive Wedelia were smaller than those between the P content and the biomass of native Wedelia, indicating that the growth of native Wedelia is dependent on phosphorus nutrition while the invasive Wedelia is more tolerant to low phosphorus nutrient. We also discussed more about this in the discussion part (Line 274-276).

Zhao, X.; Lyu, Y.; Jin, K.; Lambers, H.; Shen, J. Leaf phosphorus concentration regulates the development of cluster roots and exudation of carboxylates in Macadamia integrifolia. Front Plant Sci 2021, 11, 610591. https://doi.org/10.3389/fpls.2020.610591.

Yang, Q.; Li, Q.; Zhang, J.; Xiao, W.; Song, X. Phosphorus addition increases aboveground biomass but does not change N:P stoichiometry of chinese fir (Cunninghamia lanceolata) seedlings under nitrogen deposition. Pol J Environ Stud 2021, 30, 1421-1431. https://doi.org/10.15244/pjoes/124226.